# Sixteen Weeks of Supplementation with a Nutritional Quantity of a Diversity of Polyphenols from Foodstuff Extracts Improves the Health-Related Quality of Life of Overweight and Obese Volunteers: A Randomized, Double-Blind, Parallel Clinical Trial

**DOI:** 10.3390/nu13020492

**Published:** 2021-02-02

**Authors:** Cindy Romain, Linda H. Chung, Elena Marín-Cascales, Jacobo A. Rubio-Arias, Sylvie Gaillet, Caroline Laurent, Juana María Morillas-Ruiz, Alejandro Martínez-Rodriguez, Pedro Emilio Alcaraz, Julien Cases

**Affiliations:** 1Innovation and Scientific Affairs, Fytexia, 34350 Vendres, France; cromain@fytexia.com; 2Research Center for High Performance Sport, Catholic University of Murcia, 30107 Murcia, Spain; lhchung@ucam.edu (L.H.C.); emarin@ucam.edu (E.M.-C.); jararias@ucam.edu (J.A.R.-A.); amrodriguez@ucam.edu (A.M.-R.); palcaraz@ucam.edu (P.E.A.); 3UMR 204 Nutripass, Research Institute for Development, University of Montpellier, 34095 Montpellier, France; sylvie.gaillet-foulon@univ-montp2.fr (S.G.); caroline.laurent@univ-montp2.fr (C.L.); 4Department of Food and Nutrition Technology, Catholic University of Murcia, 30107 Murcia, Spain; jmmorillas@ucam.edu

**Keywords:** health-related quality of life, vitality, body composition, phenolic compounds, Mediterranean diet

## Abstract

Overweight and obesity adversely affect health-related quality of life (HRQOL) through day-to-day impairments of both mental and physical functioning. It is assumed that polyphenols within the Mediterranean diet may contribute to improving HRQOL. This investigation aimed at studying the effects of a polyphenol-rich ingredient on HRQOL in overweight and obese but otherwise healthy individuals. A randomized, double-blind, placebo-controlled study including 72 volunteers was conducted. Subjects were randomly assigned to receive for a 16-week period either 900 mg/day of the supplement or a placebo. Dietary recommendations were individually determined and intakes were recorded. Daily physical mobility was also monitored. Improvement of HRQOL was set as the primary outcome and assessed at baseline and at the end of the investigation using the Short-Form 36 (SF-36) health survey. Body composition was analyzed using dual-energy X-ray absorptiometry (DXA). Physical activity was calculated using the International Physical Activity Questionnaire (IPAQ). After 16 weeks, despite there being no adherence to the Mediterranean Diet Serving Score (MDSS), supplemented individuals experienced significant HRQOL improvement (+5.3%; *p* = 0.001), including enhanced perceived physical (+11.2%; *p* = 0.002) and mental health (+4.1%; *p* = 0.021) components, with bodily pain, vitality, and general health being the greatest contributors. Body fat mass significantly decreased (−1.2 kg; *p* = 0.033), mainly within the trunk area (−1.0 kg; *p* = 0.002). Engagement in physical activity significantly increased (+1308 Met-min (Metabolic Equivalent Task minutes)/week; *p* = 0.050). Hence, chronic supplementation with nutritional diversity and dosing of a Mediterranean diet-inspired, polyphenol-rich ingredient resulted in significant amelioration in both perceived physical and mental health, concomitant with the improvement of body composition, in healthy subjects with excessive adiposity.

## 1. Introduction

The World Health Organization (WHO) defines quality of life (QOL) as “an individual’s perception of their position in life in the context of culture and value systems in which they live, and in relation to their goals, expectations, standards, and concerns” [1]. Such conception of QOL is subjective, multidimensional, and encompasses a broad range of life domains, among which health is one of the most important determinants.

In addition to being an objective medical evaluation, the health-related quality of life (HRQOL), which typically combines physical, psychological, and social domains of health [2], is more recognized as a central outcome in healthcare strategies, highlighting the fact that health is deeply intertwined with a patient’s perspective. Accordingly, the use of HRQOL assessment is particularly relevant and is increasingly widespread in clinical practices [3], predominantly for ageing populations and for the inherent expansion in the prevalence of non-communicable diseases (NCDs), which include cardiovascular diseases, diabetes mellitus, and hypertension. NCDs are currently the leading cause of mortality in the modern world, contributing to 38 million deaths each year [4]. According to WHO, the global burden of NCDs is imputable to ageing, rapid urbanization, and to globalization of unhealthy lifestyles.

Among modifiable risk factors, overweight and obesity are significant contributors to the high prevalence rate of NCDs, making them the priority target in various public health programs [5]. A major cause of overweight and obesity is the excessive accumulation of body fat due to an imbalance between energy consumption and expenditure, particularly in populations with sedentary behaviors. The accumulation of body fat, predominantly within the abdominal region, is clearly associated with a chronic low-grade inflammatory state and an impaired redox status [6], both being principal pathological pathways involved in the development of NCDs. It is noteworthy that such disorders are also identified as validated metabolic indicators of ageing [7], suggesting that overweight and obesity are conditions that could potentially accelerate ageing and exacerbate the risk of ageing-linked NCDs [8].

In addition to their role in the etiology of these common medical conditions, overweight and obesity have profound adverse physical, social, and psychological consequences that can negatively affect HRQOL and impair everyday life [9,10], which appears to be an increasingly important outcome for patients and clinicians, as well as for policymakers. As a result, QOL has become a valuable endpoint assessed in both epidemiological and interventional weight management studies in overweight and obese volunteers [11,12,13]. Measuring HRQOL is the subjective perception of one’s health, where patients are asked to rate several aspects of their life. Numerous studies have demonstrated that self-rated health measures are important predictors of mortality in various populations [14].

The Short-Form 36 Health survey (SF-36), a commonly used measure in HRQOL research, is a generic methodology developed and validated in the Medical Outcomes Study [15] that assesses eight important HRQOL domains that encompass health-related social, physical, and mental dimensions. The reliability of this questionnaire has been validated both in a healthy population [16] and among people with chronic and acute health conditions, but also when comparing between different groups of patients [10]. Furthermore, several authors have demonstrated a negative correlation between BMI and SF-36 scores [17,18], and also an improvement of HRQOL correlated with weight loss in overweight people [13].

Among the dietary patterns, which are mostly studied for their health effects, it appears that adherence to the Mediterranean diet has been correlated to a lower risk for NCDs [19]; it is assumed that particular bioactive constituents of the Mediterranean diet, namely polyphenols, significantly contribute to the reported health-promoted effects [20]. Moreover, along with their widely studied antioxidant properties, recent studies demonstrate the modulatory effects of phenolic compounds on various cellular signaling pathways and responses, such as inflammation and energy metabolism [21,22], highlighting the complex mode of their individual mechanism of action in preventing NCDs [23]. Emerging large cohort studies that investigated the Mediterranean diet, specifically the regular consumption of a diversity of its main polyphenol content, have shown a positive correlation to HRQOL [24,25], suggesting that such bioactive constituents might be beneficial in improving overweight- and obesity-linked impaired HRQOL.

The aim of this clinical study was to investigate whether 16 weeks of supplementation with an accurately characterized ingredient formulated from extracts of certain fruits and vegetables commonly consumed within the Mediterranean diet could improve HRQOL in overweight and obese, but otherwise healthy, subjects.

## 2. Materials and Methods

### 2.1. Subjects

Ninety-two healthy overweight and obese subjects were recruited through advertisements in the region of Murcia in southern Spain. Both men and women between the ages of 25–55 years, being overweight to obese (BMI: 25–40 kg/m^2^) but otherwise healthy, were included in the study. Subjects were excluded if they: had a metabolic or chronic disease; had an allergy to carrot, grape, grapefruit, green tea, caffeine, or to guarana; were involved at the time of recruitment or within the previous 6 months in a chronic supplementation program, engaged in smoking cessation, or had high alcohol consumption; were pregnant or were breastfeeding; were in menopause; were suffering depression; or were involved in physical activity more than twice a week.

The study was approved by the Universidad Católica San Antonio de Murcia (UCAM), Spain) Ethics Committee (approval N° 5551) and conducted per the guidelines laid out in the Declaration of Helsinki [26] and in compliance with Good Clinical Practices defined in the ICH Harmonized Tripartite Guideline [27]. All participants were informed about the study procedures and signed written informed consent before entering the study. This trial was registered at clinicaltrials.gov as NCT03423719.

### 2.2. Test Supplement

Fiit-ns^®^, developed by FYTEXIA (France), is principally obtained by alcohol and water extraction of grapefruit (*Citrus paradisi* Macfad), grape (*Vitis vinifera* L.), and guarana seed (*Paullinia cupana* Kunth); by water extraction of green tea (*Camellia sinensis* L. Kuntze) and black carrot (*Daucus carota* L.). Fiit-ns^®^ provides bioactive compounds, specifically polyphenols from the flavonoid family, and natural components of the methylxanthine family to from an extract of guarana seeds, as well as vitamin B3. The placebo product was 100% maltodextrin, which is polyphenol-, methylxanthine- and vitamin B3-free. Both Fiit-ns^®^ and placebo were supplied in 450 mg capsules of identical appearance and flavor.

The supplement was analyzed by means of high-performance liquid chromatography (HPLC). An Agilent HPLC 1260 apparatus (Agilent Technologies, Les Ulis, France) using software Openlab CDS Chemstation Edition (version 1.3.1) coupled with a diode array detector was used. Separations were carried out by means of a Zorbax Stablebond SB-C18 column (4.6 × 150 mm; 5 µm particle size). To detect different phenolic classes, two different analytical methods were adopted: one for bioflavonoids and caffeine and one for anthocyanins.

For flavonoids and caffeine, mobile phase A consisted of water, mobile phase B was acetic acid, and mobile phase C was 100% acetonitrile. The linear gradient program was used as follows: (a) 0 to 5 min 94% A and 6% B; (b) 5 to 10 min 82.4%A, 5.6% B, and 12% C; (c) 10 to 15 min 76.6% A, 5.4% B, and 18% C; (d) 15 to 25 min 67.9% A, 5.1% B, and 27% C; (e) 25 to 30 min 65% A, 5% B, and 30% C; (f) 30 to 35 100% C; (g) 35 to 40 min 100% C; (h) 40 to 45 min 64% A and 6% B. Monitoring was performed at 280 nm at a flow rate of 1 mL/min and injection volume of 25 µL. Flavanones, flavanols, and caffeine were respectively expressed as naringin, catechin, and caffeine.

Regarding anthocyanins, mobile phase A was water, mobile phase B consisted of formic acid, and mobile phase C was acetonitrile. The gradient program used is described as follows: (a) 0 to 5 min 84.18% A, 10% B, and 5.82% C; (b) 5 to 20 min 77.6% A, 10% B, and 12.4% C; (c) 20 to 35 min 68.2% A, 10% B, and 21.8% C; (d) 35 to 40 min 58.8% A, 10% B, and 31.2% C; (e) 40 to 45 min 44.7% A, 10% B, and 45.3% C; (f) 45 to 50 min 44.7% A, 10% B, and 45.3% C; (g) 50 to 60 min 40% A, 10% B, and 50% C; (h) 60 to 65 min 84.18% A, 10% B, and 5.82% C. Monitoring was performed at 520 nm at a flow rate of 0.8 mL/min and injection volume of 10 µL. Anthocyanins were expressed as cyanidin 3-O-glucoside equivalents.

Naringin, catechin, and caffeine standards were purchased from Sigma-Aldrich Co. (St. Louis, MO, USA) and cyanidin 3-O-glucoside standard was purchased from Extrasynthese (Genay, France).

### 2.3. Study Design and Interventions

The study was designed as a 16 week, randomized, double-blinded, placebo-controlled clinical trial. Eligible participants were randomized using a simple block randomization of 1:1 with an additional stratification for sex (40% minimum and 60% maximum each sex), with a separated randomization list using computer-generated random numbers. Allocation concealment was achieved with sealed opaque envelops. Once enrolled, subjects received either the supplement (*n* = 43) or a visually identical placebo (*n* = 49). They were instructed to take two capsules daily for 16 weeks, one in the morning at breakfast and one at lunchtime.

Throughout the course of the study, volunteers were instructed by a dietitian to consume a normal caloric and balanced diet corresponding to their individual needs by determining their specific resting energy expenditure (REE), calculated from the revised Harris–Benedict equation and adjusted per individual level of physical activity [28]. At baseline (W_1_), volunteers performed a 24 h diet recall interview corresponding to the consumption of two days during the previous week and one day during the previous weekend, in order to evaluate their usual dietary habits. At the end of the studied period, the same interview was performed to check compliance with dietary instructions. A difference of ±10% between the reported and recommended intakes at the end of the study was considered as satisfactory. Moreover, general adherence to the Mediterranean dietary pattern was assessed using the Mediterranean Diet Serving Score (MDSS) [29]. This score ranges from 0 to 24, with an optimal cut-off point of 13.5, which discriminates adherent and non-adherent individuals.

Subjects were also encouraged to maintain their usual level of physical activity throughout the 16-week-long intervention period. The subjects were provided with a pedometer (HJ-321, Omron Healthcare), which was worn at the hip, to record the physical mobility as the number of daily steps. Subjects reported their daily level of activity in a diary.

Subjects reported to the UCAM Research Center for 5 visits: (i) pre-inclusion visit at week 0 (W_0_) to verify the subject’s eligibility, to assess anthropometrics, and to collect blood samples for the evaluation of safety parameters; (ii) baseline visit (W_1_); (iii) follow-up visits (W_4_, W_8_, and W_12_); and final visit (W_16_). During each visit, subjects returned their physical activity diary and the unused investigational supplements and were questioned about possible occurrence of adverse events before they were provided with a new pill dispenser for the 4 following weeks.

### 2.4. Measurements

#### 2.4.1. Health-Related Quality of Life

HRQOL was measured at baseline (W_1_) and at the end of the intervention period (W_16_) using the Spanish version of the 36-item Short Form (SF-36) health survey [30]. This generic instrument assesses participants’ self-reported HRQOL across physical and mental components. Questions pertained to the individuals’ typical day during the past four weeks and usual experiences. The 36 questions were distributed across eight subscales: physical function (PF), role-physical (RP) limitations caused by physical problems, bodily pain (BP), general health (GH) perception, vitality (VT), social functioning (SF), role-emotional (RE) limitations caused by emotional problems, and emotional well-being (EWB). The eight dimensions ranged in score from 0 to 100, with higher scores indicating better HRQOL. The SF-36 also included one Physical Component Summary (PCS) score and one Mental Component Summary (MCS) score, as well as an overall score of quality of life.

#### 2.4.2. Body Composition

At baseline (W_1_) and at the end of the study period (W_16_), body composition was assessed in the morning, with volunteers in a fasted state and wearing light clothing and no shoes.

Body weight (kg) was measured with calibrated weighing scales (TBF-300MA, Tanita Corporation, IL, USA). Waist circumference (cm) was measured at the narrowest point between the lowest rib and the iliac crest using a non-stretchable tape. The Index of Central Obesity (ICO) scores were calculated as the waist-to-height ratio.

Body fat mass was determined using a dual-energy X-ray absorptiometry (DXA) scan of the whole body (XR-46; Norland Corp., Fort Atkinson, WI, USA). Discrimination of whole-body fat mass (FM) and body trunk fat mass (TFM) was performed with a computerized software (Software Illuminatus DXA v.4.4.0, Visual MED, Inc., Charlotte, NC, USA and Norland CooperSurgical Company, Minneapolis, MN, USA) using standardized procedures.

#### 2.4.3. Self-Reported Physical Activity

The self-reported International Physical Activity Questionnaire (IPAQ) instrument was used to determine global physical activity levels [31]. This self-administered, long-form questionnaire consisted of 27 items that covered four different domains of physical activity (working, transportation, housework, and gardening and leisure-time) that occurred during the previous seven days. The results are presented as an estimation of energy expenditure in metabolic equivalent minutes per week (Met-min/week), and a categorical score was calculated to classify volunteers as inactive, moderately, or highly active. Volunteers completed the IPAQ questionnaire in the presence of an investigator at W_1_ and W_16_.

#### 2.4.4. Safety Parameters

Safety parameters were assessed before inclusion into the study (W_0_) and at the end of the intervention period (W_16_) in order to verify and confirm the healthiness of the volunteers. Safety parameters included liver function parameters (alanine transaminase (ALT), aspartate aminotransferase (AST), gamma-glutamyltransferase (GGT)), renal function parameters (urea, creatinine, sodium (Na), potassium (K)), and heart rate.

### 2.5. Statistical Analysis

Data sets were analyzed using XLSTAT-Biomed software (v. 2017.6 for Mac, Addinsoft, Paris, France). The data are expressed as the mean ± standard deviation (SD). At baseline, the distribution was considered normal. Changes within and between groups at W_1_ and W_16_ were analyzed using paired and unpaired Student’s *t*-test, respectively. To compare baseline differences between the SF-36 scales and population norms, one sample *t-*test was used. A minimum value of *p* ≤ 0.05 was selected as the threshold for statistical significance.

The primary outcome addressed in this study was the difference in SF-36 total scores after the 16 week intervention period. The power calculation was based on the previous results of a pilot study conducted with Fiit-ns^®^ [32] (α = 0.05, power (1−β) = 0.8) and was performed based on an expected clinical difference in SF-36 total scores between W_1_ and W_16_ within the supplemented group of a +5% benefit minimum to determine the targeted final sample size (*n* = 28 per group). Considering a drop-out rate of 20% and failure rate risk of 20%, inclusion of 92 subjects was recommended.

## 3. Results

### 3.1. Characterization of the Supplement

The total bioactive content corresponds to 29.27 g/100 g dry matter, with a total flavonoid content measured at 24.75 g/100 g. The flavanol content corresponded to 15.67 g/100 g and included catechin, epigallocatechin gallate, epicatechin, and epicatechin gallate, respectively, measured at 1.47, 9.55, 2.37, and 2.28 g/100 g. The flavanone content corresponded to 8.91 g /100 g, among which isonaringin, naringin, hesperidin, and neohesperidin contents were 0.54, 7.65, 0.03, and 0.13 g/100 g, respectively, whereas total unidentified flavanone was evaluated as the naringin equivalent at 0.56 g/100 g. The total anthocyanin content corresponded to 0.17 g/100 g as the kuromanin equivalent. The caffeine content was measured at 4.52 g/100 g and a third-party laboratory measured the vitamin B3 content at 2.02 g/100 g (Table 1).

### 3.2. Baseline Characteristics

From the 92 individuals who were randomly allocated to either the supplement (*n* = 43) or the placebo (*n* = 49), 78 subjects completed the 16 week intervention (85% of the randomly assigned subjects); after having started the intervention, a total of 14 volunteers dropped out for personal reasons, including 6 within the supplemented group and 8 within the placebo group. Moreover, at the end of the study period, 6 subjects were excluded from final analysis because of protocol deviation, including 2 subjects within the supplemented group and 4 within the placebo group who either did not complete the SF-36 questionnaire or who were non-compliant with the protocol. Finally, 72 volunteers were included in the analysis, with 35 individuals in the supplemented and37 individuals in the placebo group (Figure 1). Baseline data of the study population are presented in Table 2. The two groups were similar with respect to age, height, body weight, and SF-36 total scores. At baseline, the placebo group had a significantly higher average BMI compared with the supplemented group.

### 3.3. Health-Related Quality of Life

Regarding the whole population at baseline (Table 3), SF-36 subscales regarding vitality, emotional well-being, and mental component scores were significantly lower than the age-specific populations norms taken from the Spanish population reference values [33]. At W_1_, placebo and supplemented groups exhibited similar SF-36 scores, including both individual domains and summary scores (Table 4). After 16 weeks of supplementation, the supplemented group experienced a significant +5.3% increase (*p* = 0.001) (Figure 2) in total SF-36 score, while no change was observed in the total score of the placebo group. The supplemented group showed statistically significant improvements in five out of eight domains of the health-related quality of life. Respective improvements were observed for the physical component summary (PCS; +11.2%, *p* = 0.002), including physical functioning (PF; +5.5%, *p* = 0.006), bodily pain (BP; +11.2%, *p* = 0.028), and general health (GH; +7.2%, *p* = 0.010), as well as for the mental component summary (MCS; +4.1%, *p* = 0.021), which included vitality (VT; +7.8%, *p* = 0.006) and emotional well-being (EWB; +5.2%, *p* = 0.021). No statistically significant changes were shown within the placebo group after the 16 week intervention.

### 3.4. Body Composition

At baseline, all fat-mass-related variables (FM, TFM, Index of Central Obesity (ICO), and BMI) were significantly higher in the placebo group (Table 5). Such a discrepancy is explained by the higher number of obese individuals that completed the clinical investigation in the placebo group. After 16 weeks of supplementation, volunteers from the placebo group did not experience any significant changes in body composition. The supplemented group showed an improvement in anthropometrics after 16 weeks, with a statistically significant decrease in body weight by −1.3 kg (*p* = 0.013) and in BMI by −0.4 points (*p* = 0.012). Waist size significantly decreased by −1.1 cm (*p* = 0.017), consequently lowering the ICO by −1.3% (*p* = 0.018). Supplemented volunteers significantly lost −1.2 kg of FM (*p* = 0.033), of which −1.0 kg was fat lost only from the TFM (*p* = 0.002).

### 3.5. Self-Reported Physical Activity and Average Daily Steps Recording

At baseline, both groups showed similar self-reported levels of physical activity. While it did not significantly change within the placebo population (*p* = 0.280), the supplemented subjects showed an increase of +1308 Met-min/week (*p* = 0.05) after 16 weeks of supplementation (Table 6). Regarding categorical scores at baseline, the rates of volunteers within each category (i.e., inactive, moderately active, and highly active) were similar between groups. After 16 weeks, the rates of inactive people remained the same in both groups; within the placebo group, the rate of highly active subjects decreased by −43%, while the rate of moderately active individuals increased by +14%. In contrast, within the supplemented population, the rate of moderately active subjects decreased by −14% but the number of highly active individuals increased by +43%. The number of average daily steps was significantly different at baseline between placebo and supplemented subjects (*p* = 0.028). The placebo group did not experience any significant change in average daily steps monitored after 16 weeks, while the supplemented subjects significantly decreased their average rate by −678 steps (*p* = 0.019) to reach a similar level to the placebo population.

### 3.6. Recommended and Reported Dietary Intake

Recommended intake at baseline did not differ between the two groups (*p* = 0.770) (Table 7). When recommended intake was compared with reported intake at baseline, the differences were −13.7% and −7.8% for the placebo and supplemented groups, respectively. After 16 weeks, the differences between recommended and reported intake in both groups were lower than 10% (−8.8% and −9.0% for the placebo and for the supplemented groups, respectively). Mediterranean Diet Serving Scores (MDSS) were similar between both groups at 8.4 and 8.6 for placebo and supplemented populations, respectively, indicating a non-adherence to the Mediterranean diet pattern during the intervention period.

### 3.7. Safety

After 16 weeks, both liver and renal function parameters were within the healthy range in both groups, suggesting that no health impairment occurred throughout the course of the study. Moreover, heart rates stayed stable throughout the course of the study (Table 8). No adverse events or side effects linked to the supplement were reported during the course of the study.

## 4. Discussion

The main results of this study demonstrate that a 16-week-long supplementation period with an ingredient formulated from a blend of various botanical extracts, which are rich in a diversity of polyphenols and usually consumed as part of the typical Mediterranean diet, is associated with significant improvements of both the physical and mental components of the HRQOL in overweight and obese but otherwise healthy subjects of both sex.

At baseline, volunteers showed an impaired HRQOL, namely in vitality and emotional well-being subscales, for which values were below the Spanish age-specific population reference norms [33]. Although similar studies have previously reported impairment across all off the SF-36 subscales, most of them were conducted either with a population displaying a significantly higher grade of obesity or with an additional manifestation of comorbidities [10,17]. Here, baseline impairments observed for vitality and emotional well-being are in line with the work of Blissmer et al. [11], who found similar decrements in a highly comparable population of healthy overweight and moderately obese subjects, indicating higher feelings of tiredness and anxiety.

Following a 16 week intervention period associated with a normal caloric diet, both the physical and mental components of the HRQOL significantly improved in volunteers supplemented with the polyphenol-rich ingredient compared to the placebo group. Net improvements were shown by subjective ameliorations in bodily pain > general health > vitality > physical functioning > emotional well-being. It is noteworthy that after the 16-week-long period of supplementation, both the vitality and the emotional well-being values improved to achieve the level of the reference norms of the Spanish age-specific population. Improvements in these different subscales and in both the physical and mental component scores must be considered as clinically significant, as it has been stated that absolute differences of 3–5 points are clinically relevant [34].

In addition to these improvements, the 16 week chronic polyphenolic supplementation induced significant body weight loss, with an average difference between both groups of 1.1 kg. It is noteworthy that this decrease was essentially driven by an 86% fat mass reduction, for which 89% was located within the trunk area, pointing out a particularly beneficial effect on body composition. Such an improvement may, to some extent, positively impact the HRQOL. Indeed, some authors have demonstrated that weight loss was associated with improvement of both physical and mental health dimensions in several intervention trials [11,13,35,36,37]. Moreover, the amount of weight loss and the level of HRQOL improvement may be directly interconnected [37,38]. Nevertheless, here we did not demonstrate a significant correlation between weight or fat loss and HRQOL improvement, hypothesizing that weight loss could be an indirect consequence of HRQOL improvement, as it has been recently demonstrated with a bi-directional relationship between both parameters [39]. Moreover, catechins from green tea have previously been demonstrated to have antiobesity effects [40] through various mechanisms of action, such as the inhibition of pancreatic lipase [41], as well as through the regulation of obesity-related genes and proteins [42]. However, it is important to highlight that these interventional studies used significantly higher amounts of green tea catechins, whereas in the current supplement it only corresponded to one cup of green tea daily. Moreover, caffeine content and flavanones from grapefruit extract could also potentiate the decrease in body fat mass, as enhanced lipolysis leading to decreases in body weight and fat mass has previously been demonstrated in overweight and obese subjects supplemented with such kinds of bioactive compounds [43]. Accordingly, as each of the bioactive components in the supplement are present at lower levels compared to efficient dosages from the literature, it could be assumed that the beneficial observations for the supplement should be attributed to the whole formulation.

In parallel to body composition improvement, the level of physical activity, as assessed through the IPAQ questionnaire, significantly intensified (*p* = 0.05) after 16 weeks of supplementation. Thus, while 69% of volunteers from the supplemented group maintained their usual level of physical activity, 25% moved into a higher category compared with the placebo population, for which only 9% of volunteers improved their level of physical activity. Contradictorily, at the same time, the supplemented group showed a decrease in daily steps as assessed with a pedometer, while no change was observed in the placebo population. This discrepancy may be explained by the fact that pedometers are not suitable for the measurement of certain types of physical activity, such as swimming, cycling, or heavy lifting, which are otherwise assessed through the IPAQ questionnaire, making these both subjective and objective measurements, two complementary tools in physical activity assessment. As volunteers were encouraged to maintain their usual physical activity level throughout the course of the study, it can be hypothesized that the significantly higher physically active lifestyle reported within the supplemented group is not the result of conditioned mental engagement only. Indeed, the increase of HRQOL, and namely of the feeling of increased vitality, may explain such a rise in physical activity. A recent review that aimed at examining the link between physical activity and HRQOL concluded that there is a consistent cross-sectional association between physical activity level and HRQOL, namely in the vitality and in the physical functioning domains, however the finding could not confirm a causal relationship, i.e., “higher HRQOL leading to a higher level of physical activity, or vice versa, or mutual influence” [44]. Nevertheless, the engagement in a more active lifestyle within the supplemented population may also have a positive effect on body composition improvement, as discussed above.

Besides positive effects on body composition and engagement in physical activity, it appears that phenolic compounds may induce, through other various mechanisms, observable effects in terms of HRQOL improvement. Accordingly, adherence to a Mediterranean dietary pattern, characterized by wide consumption of fruits and vegetables, cereals, fish, olive oil, and red wine, has been directly associated with better QOL in an analysis including more than 11,000 participants that belonged to the SUN (Seguimiento University of Navarra) cohort [45,46]. While several nutrients and micronutrients may contribute to this effect, phenolic compounds have been suggested to be the main mediators; a large cross-sectional study demonstrated a direct relationship between the antioxidant contents of the Mediterranean diet, including the flavonoid content, and HRQOL [25]. In addition, in another recent study including more than 13,000 women, higher flavonoid intake at midlife was associated with increased odds of healthy ageing, based on higher survival at older ages free of chronic diseases and maintenance of midlife HRQOL (as assessed by the SF-36 survey) [47]. Here, despite the studied population being Spanish and particularly prone to complying with the Mediterranean diet, the MDSS did not demonstrate any significant adherence to this pattern in either groups, for whom the consumption of fruits and vegetables, the main sources of flavonoids, was below the recommendations of the last updated version of the Mediterranean Diet Pyramid [48]. Thus, it can be hypothesized that a regular basic diet has no or only a minor impact on HRQOL, since there were no improvements within the placebo population, whereas the supplemented subjects that covered the gap of phenolic micronutrients significantly improved their HRQOL.

Bioactive compounds occurring in the supplement may positively impact physiological functions related to both physical and mental health status—mainly vascular inflammation, coagulation factors, and endothelial function [25], which are all described to be impaired during overweight and obesity [49]. The aptitude of certain phenolic compounds in improving vascular health has been demonstrated both in vitro and in vivo [50]. Catechins from green tea positively impact vascular function through various complementary mechanisms linked to their antioxidative and anti-inflammatory properties, as well as to their capacity to activate endothelial NO synthase [51]. Similarly, grape polyphenols also demonstrated an aptitude to improving vascular impairments through similar molecular mechanisms [52], all contributing to a better peripheral and central blood flow, which in turn may positively affect physical and mental health status [53].

While modulation of both oxidative stress and inflammatory parameters, the main contributors in the improvement of vascular function and blood flow, has previously been demonstrated with the current supplement in a study involving obese subjects [32], specific mechanisms of flow-mediated dilation improvements and subsequent blood flow increase have not yet been investigated. Moreover, as the beneficial effect of the supplement on HRQOL has been demonstrated, further investigations will have to be conducted in attempts to confirm the causal relationship between the bioavailability and pharmacokinetics of the polyphenols metabolites and the mechanisms involved in improving vascular function.

Beyond the mentioned limitations, the results of the present study reveal the beneficial and systemic effects of phenolic compounds on subjective physical and mental symptoms linked to overweight and obesity. The study was designed to minimize bias, and thus individualized calorie intake recommendations and diet interviews, as well as monitoring of daily steps, were identified as possible confounding factors. Despite the studied population being recruited in a Mediterranean region, neither of the two groups adhered to the typical regional diet, which strengthens the hypothesis that phenolic compounds certainly contribute to subjective health, as previously proposed by others [25,47].

In conclusion, this study demonstrated that the 16-week-long consumption of an ingredient obtained from polyphenol-rich fruit and vegetable extracts associated with both caffein and vitamin B3 supports improvements in HRQOL, specifically in both mental and physical subjective feelings. In addition, the decrease in body fat mass and the significantly increased engagement in physical activity probably established a virtuous cycle between body composition, physical activity, and perceived HRQOL. The mechanisms of action likely involve improvements in vascular function via well-known antioxidative and anti-inflammatory properties of phenolic compounds. Such beneficial effects may be extended to other situations where HRQOL is impaired, particularly during the ageing process, where an imbalance of body composition and a loss of vitality and of physical functioning associated with a more sedentary lifestyle are commonly observed.

## Figures and Tables

**Figure 1 nutrients-13-00492-f001:**
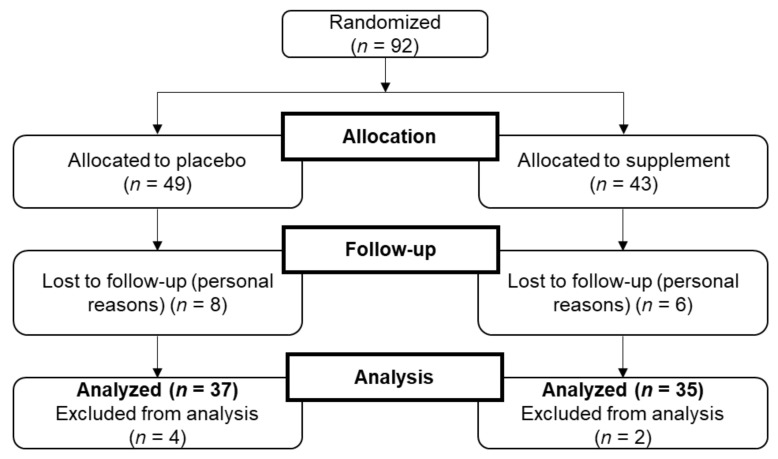
CONSORT (Consolidated Standards of Reporting Trials) flow diagram of study.

**Figure 2 nutrients-13-00492-f002:**
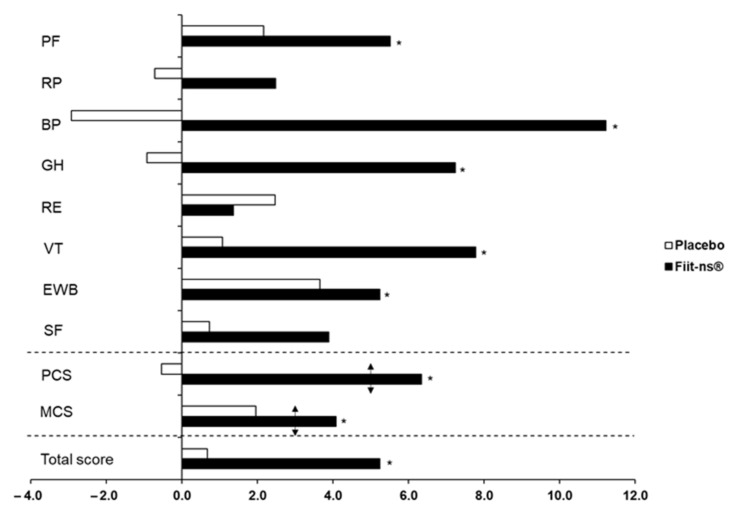
The percentage changes from baseline (W_1_) to the end of the study (W_16_) in individual SF-36 domains, as well as in the Physical Component Score (PCS), Mental Component Score (MCS), and total HRQOL score in placebo and in supplemented subjects. Arrows indicate clinically significant differences according to Samsa et al. [31]. Note: *—Indicates an intragroup difference between baseline (W_1_) and end of the study (W_16_) at *p ≤* 0.05 level. PF—physical functioning; RP—role physical; BP—bodily pain; GH—general health; RE—role emotional; VT—vitality; EWB—emotional well-being; SF—social functioning.

**Table 1 nutrients-13-00492-t001:** Characterization of phenolic compounds and caffeine present in the supplement.

Compound	Rt (min)	Λ Max (nm)	Content (mg/100 g)	
			Mean	SD
Caffeine	13.5	273	4523.2	(450.4)
Catechin	9.0	278	1472.9	(85.1)
Epigallocatechin Gallate	11.9	274	9549.9	(282.9)
Epicatechin	12.8	280	2373.0	(416.6)
Epicatechin Gallate	15.8	278	2279.3	(372.9)
Flavanone-like 1	16.1	284/323	97.6	(27.7)
Isonaringin	18.6	284/330	540.8	(156.1)
Naringin	19.5	284/330	7646.9	(66.3)
Hesperidin	20.3	284/328	25.4	(24.3)
Neohesperidin	21.2	284/328	134.2	(86.8)
Flavanone-like 2	27.6	284/329	263.5	(4.0)
Flavanone-like 3	30.4	289/328	196.4	(15.3)
Cyanidin-3-xylosylglucosylgalactoside	6.2	516	9.5	(2.0)
Cyanidin-3-xylosylgalactoside	7.9	518	39.3	(7.4)
Cyanidin-3-sinapoylxylosylglucosylgalactoside	11.2	531	13.8	(1.5)
Cyanidin-3-feruloylxylosylglucosylgalactoside	12.4	528	106.1	(13.1)
Cyanidin-3-pcoumarylxylosylglucosylgalactoside	14.3	528	3.0	(1.6)

**Table 2 nutrients-13-00492-t002:** Baseline characteristics of the study population.

Variable	Whole Population		Placebo Group		Supplemented Group	
Subjects, *n* (M/F)	72 (34/38)		37 (16/21)		35 (18/17)	
	**Mean**	**SD**	**Mean**	**SD**	**Mean**	**SD**
Age (years)	40	(6)	41	(6)	39	(7)
Height (meters)	1.69	(0.09)	1.68	(0.08)	1.70	(0.11)
Body weight (kg)	87.0	(12.6)	89.3	(12.2)	84.5	(12.6)
BMI (kg/m^2^)	30.5	(3.5)	31.6	(4.0)	29.3 *	(2.5)
SF-36 score (points)	81.6	(9.2)	81.7	(8.2)	81.5	(10.3)

*—Significant at *p* ≤ 0.05 level; M—male; F—female; BMI—Body mass index; SF-36—Short form 36 Health survey.

**Table 3 nutrients-13-00492-t003:** Mean scores for SF-36 eight subscales at baseline in the study and in the Spanish population aged 25–54.

Scale	Baseline	Age Specific Population Norm ^§^
Physical component score	83.8	83.0
Physical functioning	89.5	91.9
Role-physical	93.8	87.5
Bodily pain	79.0	82.6
General health	73.0	72.7
Mental component score	79.4 *	82.3
Role-emotional	92.4	91.3
Vitality	62.8 *	70.2
Emotional well-being	70.6 *	74.9
Social functioning	91.8	92.9

**§**—Combined score for men and women ages 25–54 as the age-specific population norm; *—a significant difference at baseline from the age-specific population norm for the scale at *p* ≤ 0.05 level.

**Table 4 nutrients-13-00492-t004:** Mean scores for SF-36 eight subscales, including both physical and mental components, and total health-related quality of life scores at baseline (W_1_) and at the end of the study (W_16_).

Scale	Placebo Group				Supplemented Group			
	W1		W16		W1		W16	
	Mean	SD	Mean	SD	Mean	SD	Mean	SD
SF-36 Total score	81.7	(8.2)	82.2	(8.2)	81.5	(10.3)	85.8 ^a^*	(6.4)
Physical component score	84.3	(7.6)	83.8	(8.9)	83.4	(12.6)	88.7 ^a^*	(7.6)
Physical functioning	88.6	(9.4)	90.6	(8.4)	90.4	(12.7)	95.4 ^a^*	(7.0)
Role-physical	94.3	(7.3)	93.6	(8.1)	93.4	(11.6)	95.7	(6.6)
Bodily pain	80.7	(18.0)	78.3	(23.3)	77.1	(22.9)	85.8 *	(14.5)
General health	73.5	(13.3)	72.8	(11.7)	72.5	(16.0)	77.8 *	(13.7)
Mental component score	79.1	(10.8)	80.7	(10.5)	79.7	(10.7)	82.9 *	(8.0)
Role-emotional	91.2	(14.8)	93.5	(12.6)	93.6	(12.0)	94.9	(9.2)
Vitality	62.4	(11.1)	63.1	(12.2)	63.1	(13.4)	68.0 ^a^*	(11.1)
Emotional well-being	70.9	(11.1)	73.5	(10.9)	70.3	(12.9)	73.9 *	(10.0)
Social functioning	91.9	(15.9)	92.6	(14.9)	91.8	(12.1)	95.4	(9.1)

*—An intragroup difference between baseline (W_1_) and end of the study (W_16_) at *p* ≤ 0.05; ^a^—an intergroup difference at the end of the study (W_16_) at *p* ≤ 0.05.

**Table 5 nutrients-13-00492-t005:** Body weight, BMI, waist circumference, ICO, total body fat mass, and total trunk fat mass scores at baseline (W_1_) and at the end of the study (W_16_).

	Placebo Group				Supplemented Group			
	W1		W16		W1		W16	
	Mean	SD	Mean	SD	Mean	SD	Mean	SD
Body weight, kg	89.3	(12.2)	89.1	(12.5)	84.5	(12.6)	83.2 ^b^*	(12.6)
BMI, kg/m^2^	31.6	(4.0)	31.5	(3.9)	29.3 ^a^	(2.5)	28.9 ^b^*	(2.8)
Waist circumference, cm	94.2	(9.7)	94.6	(9.3)	90.7	(8.8)	89.6 ^b^*	(9.3)
ICO	0.560	(0.05)	0.563	(0.05)	0.533 ^a^	(0.04)	0.526 ^b^*	(0.04)
Total body fat mass, kg	35.2	(10.4)	35.2	(10.8)	30.5 ^a^	(7.4)	29.3 ^b^*	(7.9)
Total trunk fat mass, kg	18.1	(5.4)	18.1	(5.9)	15.7 ^a^	(4.1)	14.7 ^b^*	(4.2)

*—An intragroup difference between baseline (W_1_) and end of the study (W_16_) at *p* ≤ 0.05; ^a,b^—intergroup differences at baseline (W_1_) and at the end of the study (W_16_) at *p* ≤ 0.05.

**Table 6 nutrients-13-00492-t006:** Mean total score for self-reported physical activity (IPAQ, International Physical Activity Questionnaire) and daily number of steps (pedometer) at baseline (W_1_) and at the end of the study (W_16_).

	Placebo Group				Supplemented Group			
	W1		W16		W1		W16	
	Mean	SD	Mean	SD	Mean	SD	Mean	SD
IPAQ score (Met-min/week)	4798	(4740)	4231	(4190)	4766	(4721)	6074 *	(6631)
Inactive (%)	15.2		15.2		12.5		12.5	
Moderately active (%)	63.6		72.7		65.6		56.3	
Highly active (%)	21.2		12.1		21.8		31.2	
	Mean	SD	Mean	SD	Mean	SD	Mean	SD
Daily steps	6770	(2239)	7186	(2679)	8169 ^a^	(2797)	7491 *	(2964)

*—An intragroup difference between baseline (W_1_) and end of the study (W_16_) at *p* ≤ 0.05; ^a^—an intergroup difference at baseline (W_1_) at *p* ≤ 0.05. MET-min/week—Metabolic Equivalent Task minutes per week.

**Table 7 nutrients-13-00492-t007:** Recommended and reported dietary intake at baseline (W_1_) and at the end of the study (W_16_).

	Placebo Group				Supplemented Group			
	W1		W16		W1		W16	
	Mean	SD	Mean	SD	Mean	SD	Mean	SD
Recommended intake (Kcal)	2074	(273)	2084	(281)	2096	(360)	2039	(342)
Reported intake (Kcal)	1789	(471)	1899 *	(502)	1933	(463)	1855	(392)
	**Mean**		**SD**		**Mean**		**SD**	
MDSS score	8.4		(3.7)		8.6		(4.2)	

*—An intragroup difference between baseline (W_1_) and end of the study (W_16_) at *p* ≤ 0.05.

**Table 8 nutrients-13-00492-t008:** Clinical safety values at baseline (W_1_) and at the end of the study (W_16_).

Parameters (Normal Range)	Placebo Group				Supplemented Group			
	W1		W16		W1		W16	
	Mean	SD	Mean	SD	Mean	SD	Mean	SD
Liver function								
ALT (7–55 U/L)	21.4	(9.1)	21.0	(8.5)	25.0	(15.9)	25.1	(14.7)
AST (8–48 U/L)	20.2	(5.5)	20.1	(5.3)	22.4	(6.3)	23.3	(10.0)
GGT (6–48 U/L)	19.1	(11.7)	19.9	(11.9)	23.1	(13.4)	24.7	(14.0)
Kidney function								
Urea (15–46 mg/dL)	35.4	(9.1)	33.0	(7.7)	31.9	(7.8)	30.1	(7.4)
Creatinine (0.6–1.3 mg/dL)	0.79	(0.18)	0.74 *	(0.16)	0.74	(0.15)	0.76	(0.16)
Na (135–145 mmol/L)	141.2	(1.2)	141.2	(2.1)	141.7	(1.7)	141.1	(1.7)
K (3.6–5.2 mmol/L)	4.3	(0.3)	4.3	(0.2)	4.3	(0.2)	4.3	(0.3)
Heart rate (bpm)	71.8	(10.2)	72.1	(8.6)	71.3	(11.2)	70.4	(16.4)

*—An intragroup difference between baseline (W1) and end of the study (W_16_) at *p* ≤ 0.05.

## Data Availability

The data presented in this study are available on request from the corresponding author, due to privacy restriction.

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
