# Peer review of "Sixteen Weeks of Supplementation with a Nutritional Quantity of a Diversity of Polyphenols from Foodstuff Extracts Improves the Health-Related Quality of Life of Overweight and Obese Volunteers: A Randomized, Double-Blind, Parallel Clinical Trial"

_nutrients, 2021, doi:10.3390/nu13020492_

Round 1
Reviewer 1 Report
The manuscript by Cindy Romain et al. concerns the properties of a natural polyphenols extract. The aim of this clinical study is to investigate whether the supplementation with a preparation developed by a company in developing scientifically active nutrients could improve health-related quality of life in overweight and obese, but otherwise healthy, subjects, after 16-week consumption.
The paper is interesting and the findings are relevant.
However, there are some recommendations.
My concern is about the article title. The title is misleading and needs to be changed by eliminating the words Mediterranean diet. In the Materials and Methods section, the composition of the extract used is described: grapefruit (Citrus paradisi Macfad), grape (Vitis vinifera L.) and guarana seed (Paullinia cupana Kunth), green tea (Camellia sinensis L. Kuntze) and black carrot (Daucus carota L.) (lines 122-124). The foods used for the preparation of the extract are not all typical foods of the Mediterranean diet. Therefore it is not correct to associate the extract with the Mediterranean diet. It is correct to say "extract rich in polyphenols".
Abstract is clear and well written: it provides a general idea of the experimental design and of the results.
Overall, manuscript is clear and well organized. I recommend adding recent bibliography related to “polyphenols and obesity” in the Introduction section.
Author Response
The authors would like to thank the reviewer for its review and the relevant comments that help us improve the quality of our manuscript.
We agree that the title is quite misleading as linking the polyphenol extracts, that are at the basis of the supplement, to the Mediterranean diet is also tricky. Accordingly, we propose to modify the title for:
“16-week supplementation with a nutritional quantity of a diversity of polyphenols from foodstuff extracts improves health-related quality of life of overweight and obese volunteers: a randomized, double-blind, parallel clinical trial”
As suggested by the reviewer, we propose to add recent bibliography regarding polyphenols and obesity as followed:
Line 94: “…such as inflammation and energy metabolism [21-22] …”
[22] Boccellino, M.; D’Angelo S. Anti-obesity effects of polyphenol intake: current status and future possibilities. Int J Mol Sci 2020, 21, 5642.
Line 95: “…in preventing NCDs [23].”
[23] Castro-Barquero, S.; Lamuela-Raventós, R.M.; Doménech, M.; Estruch, R. Relationship between Mediterranean dietary polyphenol intake and obesity. Nutrients 2018, 10, 1523.
Reviewer 2 Report
- The paper is well written and everything is comprehensible.
- Lines 105-112: The exclusion criteria are the appropriate ones.
- Lines 135-150: The HPLC analyses are well organized.
- Line 150: I recommend authors to use the name: Cyanidin 3-O-glucoside instead of kuromanin.
- The size of the sample is appropriate for such a study.
- Table 2: Authors maybe should check again whether the body weight significantly differs among the supplemented and the placebo groups.
- Phenolic compounds are gaining the interest of the scientists more and more due to their health promoting properties, thus the findings of the present study are of significance regarding those micro-constituents’ activities.
- In general, this study consists of well organized protocols that are fully supported by the exported results.
Author Response
The authors would like to thank the reviewer for its review and the relevant comments that help us improve the quality of our manuscript.
Table 2: we have again performed a student T-test analysis on the raw data to determine whether the average body weight at baseline for both the placebo and the supplement groups were significantly different or not. Both groups are not significantly different, p-value = 0.120.
We agree that Cyanidin 3-O-glucoside is more common than Kuromanin, and we propose to modify the manuscript accordingly:
Line 151-153: “Anthocyanins were expressed as Cyanidin 3-O-glucoside equivalent. Naringin, catechin, and caffeine standards were purchased from Sigma-Aldrich Co. (St. Louis, M.O., U.S.A.), and Cyanidin 3-O-glucoside standard from Extrasynthese (Genay, France).”
Reviewer 3 Report
This is a well-organized study exploring the effects of supplementation with Mediterranean diet-inspired ingredients on quality of life. Overall, the work is well done, but some questions remain.
Some specific comments and questions:
INTRODUCTION
L47: Change ‘be’ to ‘being’.
RESULTS
Table 4 is a bit difficult to read. Perhaps putting the SD in parentheses may be helpful. Same for Table 5.
DISCUSSION
L366: Delete ‘healthy’ (i.e., in ‘healthy overweight’).
L399: The last section of this paragraph discusses the difficulty of attributing the observed weight/fat loss to any specific ingredient in the supplement. It should be made clear that the observed physical and mental effects of the supplement are attributable to the ingredients as a whole, and that the effects of a single isolated ingredient cannot be known. Indeed, there may be interactions between ingredients as well, which is not studied here.
L409: This introductory sentence implies cause and effect, which cannot be known here. Please rephrase.
Paragraph beginning on L409: This paragraph is problematic, as the authors attempt to explain the apparent discrepancy between physical activity assessment results and a physical measurement tool (pedometer), all within the context of HRQOL. As the authors note, according to a recent review there is no causal relationship between physical activity and HRQOL. It is difficult to assess the validity of these arguments since detailed physical activity results were not presented (swimming, cycling, etc.). It may be helpful to at least summarize some physical activity data to better explain this discrepancy, otherwise we run the risk of ignoring data not favorable to our hypothesized outcome.
L475: Unless I missed this, the confounding factors identified were never addressed statistically. Simple t-tests would not be enough. Perhaps more sophisticated statistical test are called for.
Limitations: From line 306, “At baseline, all fat mass-related variables (FM, TFM, Index of Central Obesity (ICO) 306 and BMI) were significantly higher in the placebo group.” This is important and possibly a severe limitation. The placebo group were heavier (granted, not significant) with higher obesity. This could very well have affected the outcome of this study and should be noted as an important limitation.
OVERALL
The article is certainly well written, and the work looks to be well done. The Discussion needs to be edited to make certain points clearer and to point out some obvious limitations.
Author Response
The authors would like to thank the reviewer for its review and the relevant comments that help us improve the quality of our manuscript.
Line 47: we propose to change “be” to “being”.
We agree that SD should be put in parenthesis and we have modified accordingly Table 1, 2, 4, 5, 6, 7 and 8.
Line 370: we propose to delete “healthy” as followed “… in overweight and obese…”
We agree with the comment on Line 403 stating that observed benefits on both physical and mental effects have to be attributed to the whole supplement. Indeed, the purpose of our comments that follows in the manuscript is to emphasize this statement, highlighting that despite individual botanical extracts have previously been demonstrated for benefits, however, here, in the supplement, their content is significantly lower, and the benefits are accordingly the consequence of their association in the supplement. We propose to add in the manuscript at line 412 in what follows the part in italic: “… with such kind of bioactive compounds [41]. Accordingly, as each of the bioactive of the supplement are quite lower compared to efficient dosages from the literature, it could be assumed that the beneficial observations for the supplement should be attributed to whole formulation.”
Line 413, we agree that the introductory sentence is quite misleading, and we propose to change for what follows in italic: “In parallel to body composition improvement, the level of physical activity …”
Regarding physical activity assessment through IPAQ questionnaire measurement, the only validated endpoints we can discuss are either related to activity intensity (Inactive; Moderately active; Highly active), or to physical activity category (Job domain; Transport domain; Domestic/Garden domain; Leisure time domain). Raw data of the IPAQ questionnaire do not inform us about what kind of physical activity have been performed; in the discussion of our manuscript, swimming, cycling, ang heavy lifting were just examples but do not correspond to measured endpoints.
Line 475, the possible confounding factors discussed were analyzed to evaluate possible bias with primary outcome and nor the calories ingested, neither the daily steps were, through linear regression analysis, identified to interact with SF-36 score (data not shown).
We agree that fat mass-related variables were higher in the placebo group, and a stratification could have been considered as being interesting. However, we do not consider this as a severe limitation. Indeed, we remind that regarding the primary outcome, both groups were similar at baseline, which may signify that the fat mass-related variables do not influence to such a significant extent the primary outcome.
Reviewer 4 Report
16-week supplementation with a Mediterranean diet-inspired ingredient rich in a diversity of polyphenols improves health-related quality of life of overweight and obese volunteers: a randomized, double-blind, parallel clinical trial
In a complex intervention study, the authors investigated if a polyphenol-rich ingredient could improve health-related quality of life (HRQOL) in obese and overweight participants. They concluded that a chronic supplementation with a polyphenol-rich ingredient resulted in amelioration in HRQOL.
The private supported study is well-done, the manuscript is well-written, but the conclusions are not correct.
Major problems:
- The supplement contained not only polyphenols, but also caffeine and vitamin B3.
- The weight loss effect was higher in the supplemented group which can influence the increased quality of life
- Physical activity seriously increased in the supplemented group, which can influence the increased quality of life
- No information about sample size calculation and allocation concealment
From this data, it is impossible to attribute the improved HRQOL to polyphenols.
Author Response
The authors would like to thank the reviewer for its review and the relevant comments that help us improve the quality of our manuscript.
We agree with the reviewer that the supplement contains not only polyphenols, but also caffeine and vitamin B3. However, the level of caffeine in the supplement, 40.7 mg daily, and the level of vitamin B3 in the supplement, 18.2 mg daily, respectively correspond to ½ cup of Starbuck espresso coffee (30 mL) for caffeine ingestion (Ludwig et al., Food Funct, 2014, 5, 1718-26), and to the r ange for the recommended daily intake in the USA according the National Institutes of Health (https://ods.od.nih.gov/factsheets/Niacin-HealthProfessional/) for vitamin B3 ingestion. Regarding such a dose of caffeine, as the usual consumption of coffee through various countries in Europe goes from 98 mL to 798 mL daily (Zamora-Ros et al., Int J Cancer, 2014, 135, 1470-9), it cannot reasonably be considered that the supplement provides a quantity of caffeine able to provide an additional effect to what usual coffee consumption already does. The main reason why such a dose of caffeine is formulated with polyphenols of the supplement is based on the work from Sansone et al., Am J Clin Nutr, 2017, 105, 352-60, that demonstrated low caffeine ingestion benefits on polyphenol bioavailability. Similarly, as we did not find any bibliography demonstrating that daily ingestion of recommended daily intakes of vitamin B3, or even more, is linked to improved Health-Related Quality of Life, we do not have any clue that vitamin B3 may influence the primary outcome.
We agree with the reviewer that the weight loss effect was higher in the supplemented group and we already discussed the relationship with Health-Related Quality of Life, line 396-403.
We agree with the reviewer that physical activity seriously increased in the supplemented group and we already discussed the relationship with Health-Related Quality of Life, line 424-435.
The details of the power calculation are presented line 238-244, and allocation concealment explained line 155-159.
Our demonstration does not support an improved quality of life attributable to either caffein or to vitamin B3, accordingly, we assume it must me attributable for this population to the quantity and to the diversity of polyphenol daily ingestion from the supplement. However, we propose to modify the conclusion adding line 485 what follows in italic: “In conclusion, this study demonstrated that the 16-week long consumption of an ingredient obtained from polyphenol-rich fruit and vegetables extract associated to both caffein and vitamin B3 supports improvements in HRQOL”.
Round 2
Reviewer 4 Report
From this study, it is impossible to know if the intervention has an effect on HRQOL, on physical activity and/or on body weight. Moreover, the weight loss is strange, because the authors wrote on line 163: "Throughout the course of the study, volunteers were instructed by a dietitian to consume a normal calorie and balanced diet corresponding to their individual needs".
Did the intervention improves HRQOL, and did HRQOL improves physical activity and loss of body weight? Or did physical activity improves HRQOL ...
The main problem is that there were multiple interventions during the follow-up: polyphenols, vitamins, caffeine, physical activity, weight loss; it is impossible to attribute the outcome to one element. The right conclusion of the study is that polyphenols, caffeine, vitamin B3, physical activity and weight loss will improve HRQOL.
Allocation concealment is not explained at line 155-159.
My ethical concerns are the conflicts of interest: Fytexia has a commercial interest in this publication, and too much efforts are done to sell polyphenols alone.
Author Response
The authors would like to thank the reviewer for its new review (round 2) and for the relevant comments that help us improve the quality of our manuscript.
The primary outcome of the present work has been determined as being improvement of the health-related quality of life assessed with the SF-36 health survey. The primary outcome has been measured with a validated questionnaire for which it has been calculated a statistical power to achieve +5% performance in our study. We agree that HRQOL, physical activity, and body weight, significantly improved, but here, the main question of our work is not to try establishing a causal relationship between these endpoints, it is rather to validate if whether the primary outcome improves or not after taking the supplement. Then after, during the discussion, we debate the various possible mechanisms that may have a causal relationship with improvement of the primary outcome during the intervention.
To our knowledge, it is not strange that we observe a weight loss despite the volunteers follow a normal calorie and balanced diet. Indeed, effects on the metabolism with increased energy expenditure may easily, in this extent of body weight loss, explain that. Furthermore, there is some literature demonstrating that few phenolics are able to increase cAMP-PKA pathways involved for lipolysis or for uncoupling of oxidative phosphorylation.
We do not agree with the reviewer 4 when he states that this is a multiple intervention. Indeed, we managed main possible confounding, i.e., the recommended calorie intake and the physical activity variations that were both verified to do not interact with the primary outcome improvement through linear regression analysis as formerly explained to Reviewer 3 during round 1 of our answers (available in the previous cover letter). Accordingly, we will not propose to consider these two outcomes are part of the intervention since they are managed and that they do not interfere. In addition, and based on our explanation, a proposition to modify our conclusion has already been sent in round 1 of our answers to Reviewer 4 when we proposed to modify the conclusion adding line 515 what follows in italic: “In conclusion, this study demonstrated that the 16-week long consumption of an ingredient obtained from polyphenol-rich fruit and vegetables extract associated to both caffein and vitamin B3 supports improvements in HRQOL”.
Regarding allocation concealment we propose to add line 166 what follows in italic: “… with a separated randomization list using computer-generated random numbers. Allocation concealment was achieved with sealed opaque envelops. Once enrolled …”
We may understand that someone could have doubts when there is the possibility of a conflict of interest. Nevertheless, we observed that it is not so usual that when a work is granted by a private company or even by whoever else, that conflicts of interest are declared. We declared that this work has been granted by a company in order to be fully transparent. In addition, for this intervention study we would like to highlight that the protocol has been published on clinicaltrials.gov in order to be transparent. We also inquired to pre-publish this work as preprint in order to be transparent with even more potential comments from the whole scientific community. Accordingly, we think that in our position it is difficult to provide more transparency, and we are sure that the scientific community would certainly appreciate that all the research work that are performed may be achieved following such a transparency.